# Role of the Microbiome in *Aedes* spp. Vector Competence: What Do We Know?

**DOI:** 10.3390/v15030779

**Published:** 2023-03-17

**Authors:** Qesya Rodrigues Ferreira, Fabian Fellipe Bueno Lemos, Matheus Nascimento Moura, Jéssica Oliveira de Souza Nascimento, Ana Flávia Novaes, Isadora Souza Barcelos, Larissa Alves Fernandes, Liliany Souza de Brito Amaral, Fernanda Khouri Barreto, Fabrício Freire de Melo

**Affiliations:** Instituto Multidisciplinar em Saúde, Universidade Federal da Bahia, Vitória da Conquista 45029-094, Brazil

**Keywords:** mosquito vectors, *Aedes*, microbiota, vector borne diseases, arboviruses, vector competence

## Abstract

*Aedes aegypti* and *Aedes albopictus* are the vectors of important arboviruses: dengue fever, chikungunya, Zika, and yellow fever. Female mosquitoes acquire arboviruses by feeding on the infected host blood, thus being able to transmit it to their offspring. The intrinsic ability of a vector to infect itself and transmit a pathogen is known as vector competence. Several factors influence the susceptibility of these females to be infected by these arboviruses, such as the activation of the innate immune system through the Toll, immunodeficiency (Imd), JAK-STAT pathways, and the interference of specific antiviral response pathways of RNAi. It is also believed that the presence of non-pathogenic microorganisms in the microbiota of these arthropods could influence this immune response, as it provides a baseline activation of the innate immune system, which may generate resistance against arboviruses. In addition, this microbiome has direct action against arboviruses, mainly due to the ability of *Wolbachia* spp. to block viral genome replication, added to the competition for resources within the mosquito organism. Despite major advances in the area, studies are still needed to evaluate the microbiota profiles of *Aedes* spp. and their vector competence, as well as further exploration of the individual roles of microbiome components in activating the innate immune system.

## 1. Introduction

Arthropod-borne virus (arbovirus) is a non-taxonomic designation for viruses that can replicate in both vertebrate hosts and arthropod vectors [1,2]. There are >500 recognized arboviruses worldwide distributed into seven families: *Togaviridae*, *Flaviviridae*, *Bunyaviridae*, *Reoviridae*, *Rhabdoviridae*, *Orthomyxoviridae*, and *Asfarviridae*. About 150 of these, mainly from the genera *Flavivirus* spp. and *Alphavirus* spp., are known to cause severe human illness [3,4]. The most prevalent arboviral diseases in the last four decades—namely, dengue fever (DENV; 100 million symptomatic cases/year), chikungunya virus (CHIKV; 693,000 cases/year), Zika virus (ZIKV; 500,000 cases/year), and yellow fever (YFV; 130,000 cases/year)—represent an important cause of morbidity and mortality in developing countries. Accordingly, they also pose a substantial economic burden for health systems, particularly in tropical and subtropical regions [5,6].

The past 50 years have witnessed a dramatic re-emergence of epidemic arboviral diseases, mostly resulting from the modern world triad: urbanization, globalization, and international mobility [7,8]. Re-emergence is not an exclusive characteristic of arboviruses; however, their rapid and geographically extensive dispersal constitutes an important difference from other pathogens [9]. The ongoing geographical expansion of the DENV pandemic in the tropics and subtropics, along with the explosive onsets of CHIKV and, more recently, ZIKV infection-associated neurological disorders and neonatal malformations in Latin America, have all highlighted the need for an active fight to prevent the viral perpetuation and the emergence of outbreaks [10,11,12].

Arboviruses usually have a short incubation period ranging from 3 to 10 days, and most infected individuals remain asymptomatic or present mild symptoms [13]. Dengue is a potentially more serious infection due to its hemorrhagic component that can lead to circulatory collapse [13]. This disease is currently present in more than 100 tropical and subtropical countries, with 40% of the world’s population at risk of acquiring the disease [14]. The 2014 Zika epidemic stands out, a period in which several countries, mainly in South America, were affected [15]. Zika disease is characterized by arthralgia, myalgia, rash, headache, fever, and enlarged lymph nodes and is usually self-limiting. However, it has complications such as the demyelinating polyneuropathy Guillain-Barré Syndrome and Congenital Zika Syndrome, the only virus capable of such in the *Flaviviridae* family [13]. Chikungunya is an acute infection in which the main symptoms are high fever and severe polyarthralgia. It has a mortality rate of about 0.1%, and the elderly population is the most affected. The disease can also manifest itself through more protracted symptoms, such as chronic polyarthralgia, neuropathic pain and mood swings, which can significantly impair the patient’s quality of life. CHIKV virus can be transmitted vertically during birth, resulting in congenital encephalitis in half of the cases [13,16].

Due to the lack of vaccines against arboviruses licensed at this time, the battle against vector-borne diseases mostly relies on vector control [12]. DENV, CHIKV, ZIKV and YFV are predominantly transmitted horizontally to humans through the bite of infected female mosquitoes *Aedes aegypti* and *A. albopictus* in sporadic outbreaks over an interval of years or annually in a cyclical, seasonal pattern [17,18,19]. *Aedes aegypti* are small, dark mosquitoes with white markings on their legs and a wide distribution worldwide, especially in tropical and subtropical regions with environmental disturbances [20,21]. On the other hand, the likely secondary vector of arboviruses, *A. albopictus*, are known to colonize all five continents [22]. Despite this extensive geographic distribution, the transmission of mosquito-borne arboviruses requires high densities of competent vectors, a high vector survival rate, and frequent contact between vectors and susceptible vertebrate hosts [23]. When taken together, these factors constitute vector capacity, which, alongside vector competence—i.e., the intrinsic ability of a mosquito population to become infected with a particular pathogen and transmit it to susceptible hosts [24]—determines the efficiency of a vector population to transmit a pathogen under natural conditions.

In this sense, a multitude of microbes that colonize different organs and tissues of mosquitoes, including the intestine, salivary glands, and reproductive tissues, seem to play a pivotal role in the vectorial ability to transmit viral pathogens [25]. High-performance sequencing and metagenomic analyses have advanced our understanding of the composition and functionality of the microbiota of *Aedes* spp. [26,27]. Recent studies have highlighted that the mosquito microbiota can modulate the susceptibility to arboviral infection, both through the production of antiviral proteins and the activation of the mosquito’s innate immune system [28]. Dissecting these mechanisms underlying vector competence constitutes an important field in the development of effective strategies to control the spread of arboviral epidemic diseases. In this study, we aim to explore the role of the microbiome on the vector competence of *Aedes* spp. and to discuss the novelties and perspectives in this area.

## 2. Methods

This comprehensive review aimed to investigate the role of the microbiome of *Aedes* spp. in vector competence. We performed a search of PubMed/MEDLINE, Virtual Health Library (BVS), and Latin American and Caribbean Health Sciences Literature (LILACS) databases until February 2023. Medical Subject Headings (MeSH) index terms and free-text words were combined for search strategy development. Search terms included ‘Aedes [Mesh]’, ‘Microbiota [Mesh]’, ‘Microbiome’, ‘Microbial Community’, ‘Microbial Community Composition’, ‘Microbial Community Structure’, and ‘vector competence’. Boolean operators (AND, OR) were also used to narrow or broaden the search as required. The titles and abstracts of the articles were analyzed, and studies that answered the research question were included. Subsequently, we described the main findings of the retrieved studies and their limitations and pointed out prospects in the research field.

## 3. *Aedes* spp.: Understanding the Vector Interaction with Arboviruses

For mosquito-borne arboviruses, under natural conditions, a susceptible adult female becomes infected by taking a blood meal from a viremic host [28]. Upon ingestion, arboviruses must infect and replicate in midgut epithelial cells, thereby overcoming the midgut infection barrier (MIB), the host’s potent innate immune responses, finally facing the effects of the luminal and sometimes internal microbiota [29,30]. Subsequently, they must escape, disseminate and invade the salivary gland when the vector is able to transmit the arbovirus by a bite to a new host [31]. On the other hand, arboviruses can also be maternally transmitted by infected females to their offspring through the infection of germinal tissues [32].

The MIB in refractory mosquitoes could be attributed to multiple factors, including (1) a lack of adequate cell surface receptors for the pathogens to initiate infection; (2) virus filtration by peritrophic matrix; (3) its inactivation by midgut digestive enzymes; (4) strong immune responses against pathogen replication; or (5) effects of luminal and sometimes internal microbiota [33]. Indeed, the first step in the responses of *Aedes* spp. against arboviruses occurs on the surface of the midgut epithelium itself [34]. Nevertheless, pattern recognition by specialized receptors is the first step in orchestrating robust innate immune responses. Upon viral-patterns recognition, pathogen-recognition receptors (PRRs) trigger and regulate distinct downstream complex immune signaling pathways such as Toll, the immune deficiency (Imd), the Janus kinase/signal transducers, and activators of transcription (JAK-STAT), and the RNA interference (RNAi) specific antiviral response pathways [35]. After this interaction, humoral and cellular responses take place, inducing both antimicrobial peptides (AMPs) expression and hemocyte-mediated immune responses, such as phagocytosis, nodulation, and encapsulation, respectively [36,37].

The Toll pathway is recognized for its importance in the immune response to Gram-positive bacteria, fungi, and especially DENV in mosquitoes [38]. Recognition of arbovirus-related PAMPs by PRRs triggers proteolytic cascades that lead to the cleavage of the cytokine Späetzle. In a Späetzle -cleavage-dependent manner, Toll transmembrane receptor activation drives MyD88, Tube, and Pelle-mediated intracellular signaling pathway. It thereby culminates in phosphorylation and subsequent proteasomal degradation of Cactus, which binds to the NF-κB-like transcription factor Dorsal (Rel1). Finally, Cactus degradation allows the translocation of Rel1 to the nucleus and subsequent expression of AMPs and other immune effectors [39,40]. The transcription of innate immunity genes encoding AMPs is also highly IMD-dependent. In contrast to the Toll, the Imd pathway is also activated during Gram-negative bacterial infection, and its induction is known to lead to the degradation of the negative regulator Caspar. This process promotes the translocation of Relish 2 (Rel2) to the nucleus, which activates IMD-regulated AMPs transcription [41,42]. Similarly, the JAK-STAT pathway is suggested to play an essential role in antiviral defense in mosquitoes [43]. Finally, it is important to mention that, despite not being a classical immune signaling pathway, mosquito RNAi is the major innate immune pathway controlling antiviral response and, thereby, arbovirus infection and transmission. The RNAi machinery is triggered when dsRNA enters a cell and results in the degradation of cognate viral mRNA, thus inhibiting viral replication and promoting viral clearance [43,44].

These common pathways between immune responses to arboviruses and other microorganisms help to explain why increasing evidence also highlights microbiota as an important exponent of mosquito-arbovirus interactions. It appears that the presence of the microbiome induces a state of basal activation of the innate immune system of *Aedes* spp. that would be associated with modulation of susceptibility to arbovirus infection [45,46]. Figure 1 outlines this potential three-way interaction, which will be the subject of further discussion below.

## 4. Formation and Composition of the Microbiome

The formation of the microbiome begins in the early developmental stages of *Aedes* spp., since through vertical transmission, eggs inherit microorganisms from their progenitors [47]. Already in the larval stage, there is a strong interaction with the developing water, where the symbiosis occurs primarily through feeding [48,49]. Factors such as geographic location, physicochemical characteristics of the water, larval population density, and food availability are the main determinants of the microbiome in this first moment and may promote or inhibit the colonization of the larval organism [50]. Dickson et al. recently attempted to identify if habitat-related differences in bacterial communities in larval development sites could mediate environmental variation in vector-borne pathogen transmission. The authors showed that exposure to different bacteria during larval development could influence the variation in adult traits in *A. aegypti*. More interestingly, it also appears to be the potential transmission mechanism of medically relevant human pathogens [51]. In turn, Wang et al. found that the microbiota of *A. albopictus* is strongly determined by its larval stage habitat and physiological status (e.g., developmental stage) [52].

During adulthood, both intrinsic and extrinsic host factors can cause changes in the microbiota. Among the extrinsic factors, geographic location and food availability remain important influencers. Mosquitoes collected from different regions of the world show significant taxonomic variation in terms of microorganisms, suggesting that the microbiota has native characteristics, which may partly explain the differences in the incidence of arboviruses around the globe [26]. However, these differences are not as significant at the phylum level, and greater variability exists at lower taxonomic levels [48]. This may support the idea that what occurs is the sum of a “core microbiome” composed of similar Operational Taxonomic Units (OTUs), regardless of geographic location, with locally acquired microorganisms. This resulting microbiome is key to understanding and controlling the transmission of arboviruses by mosquitoes of the genus *Aedes* spp. [25].

As for the factors intrinsic to the host, we can highlight mating and feeding type. A study by Díaz et al. analyzed the reproductive tract of Aedes aegypti and Aedes albopictus females using metabarcoding of the 16S rRNA, where it was possible to observe that blood feeding and mating cause changes in the relative populations of the microbiome components in order to favor the metabolic pathways associated with egg laying [53]. In addition, it was observed that sugar fermentation increases the number of anaerobic microorganisms [48]. Oliveira et al. also demonstrated that mosquitoes fed only sugar produce a greater amount of reactive oxygen species (ROS) when compared to those fed only blood. The low availability of ROS is conducive to the expansion of bacterial levels, causing higher mortality [54]. Finally, blood consumption of hosts treated with antibiotics causes a decrease in the number of bacteria present in the microbiome. This impairs blood digestion and, consequently, egg laying by females [48].

The arthropod microbiome is composed of several microorganisms, of which we will highlight bacteria, fungi, and viruses. Other forms of microorganisms, such as helminths, have also been described [25], but further studies are needed to clarify which ones and how they act in the organism of these insects. Table 1 summarizes the results of the most important studies on the microbiome composition of *Aedes* spp.

### 4.1. Bacteria

Bacteria are widely found in various tissues of *Aedes* spp. mosquitoes and constitute more than 99% of the components of the microbiome [25]. In a recent study, Lin et al. collected mosquito samples reared under the same laboratory conditions and compared the microbial composition of the midgut and entire bodies of *A. aegypti* and *A. albopictus* from Southern China using 16S rRNA gene sequencing. The authors found that microbes in the entire bodies of both male and female *A. aegypti* mainly included seven genera: *Leptothrix* spp., *Methylobacterium* spp., *Enterobacter* spp., *Methylotenera* spp., uncultured bacteria, *Escherichia-Shigella* spp., *and Sphingomonas* spp. No microbe was absolutely dominant [56]. Conversely, the bacterial composition of *A. albopictus* is concentrated in a few genera, such as *Wolbachia* spp. and *Bacillus* spp.. Another work carried out in the Kandy district, Sri Lanka, showed that the microbiota found in the larval intestine of *A. aegypti* was composed of bacteria of the genus *Bacillariophyta* spp., *Cyanobacteria/Cyanophyta* spp., and *Ochrophyta* spp. in its majority, followed by *Charophyta* spp. and *Euglenozoa* spp. in a lower frequency. However, the midgut contents of *A. albopictus* belonged mainly to *Chlorophyta* spp., and less frequently *Cyanobacteria/Cyanophyta* spp., *Bacillariophyta* spp., *Euglenozoa* spp., *Ochrophyta* spp., and *Charophyta* spp. [65]. A survey in Panama revealed the cultivable bacterial species in the midgut of adult females of *A. aegypti* were six distinct genera: *Asaia* spp., *Aeromonas* spp., *Enterobacter* spp., *Paenibacillus* spp., *Proteus* spp., and *Comamonas* spp. [45]. In Brazil, the main bacteria isolated from the midgut of *A. aegypti* were *Acinetobacter* spp., *Aeromonas* spp., *Cedecea* spp., *Cellulosimicrobium* spp., *Elizabethkingia* spp., *Enterobacter* spp., *Lysinibacillus* spp., *Pantoea* spp., *Pseudomonas* spp., *Serratia* spp. and *Staphylococcus* spp., 72% of them Gram-negative [55]. Thus, it is possible to notice a great variability of genera that infect *Aedes* spp., despite some of these being identified in more than one study.

Indeed, the available studies on the establishment of a core midgut microbiome profile of *Aedes* spp. are somewhat divergent, even within the same species. Nascimento et al. aimed to describe the culture-dependent native microbiota associated with the female *A. aegypti* (strain PP-Campos) through 16S rRNA gene sequencing. Their results revealed eleven isolates from the native bacterial community of *A. aegypti* (strain PP-Campos)—*Acinetobacter* spp., *Aeromonas* spp., *Cedecea* spp., *Cellulosimicrobium* spp., *Elizabethkingia* spp., *Enterobacter* spp., *Lysinibacillus* spp., *Pantoea* spp., *Pseudomonas* spp., *Serratia* spp., and *Staphylococcus* spp. [55]. In another Brazilian study, David et al. aimed to profile the midgut microbial diversity throughout the *A. aegypti* lifespan and address possible determinants of microbiota structure in adult females collected in the northwest of Rio de Janeiro city (RJ). In turn, these authors suggested that *A. aegypti* harbors lifelong stable core microbiota mostly composed of the genera *Pseudomonas* spp., *Acinetobacter* spp., *Aeromonas* spp., and *Stenotrophomonas* spp. and the families *Oxalobacteraceae*, *Enterobacteriaceae*, and *Comamonadaceae* [28]. Regarding *A. albopictus*, *Rosso* et al. highlighted that *A. albopictus* collected in Italy had lower diversity and different microbiota composition compared to samples collected in France and Vietnam. Their results suggest that the mosquito possesses a core microbiota consisting mainly of the genus *Pseudomonas* spp. [59]. Minard et al. counter these findings by demonstrating that all mosquitoes collected from different sites have a bacterial microbiota dominated by a single taxon, *Wolbachia pipientis*. However, the authors note that the diversity index values likely underestimate the true diversity of the mosquito microbiota due to the high abundance of *Wolbachia* spp. sequences [63]. These studies indicate that when it comes to wild mosquitoes, the composition of adult midgut microbiota between different strains derived from distinct geographic populations of *Aedes* spp. is markedly distinct.

The bacterial composition of the microbiome changes significantly throughout the developmental stages of the mosquito. During metamorphosis, approximately 90% of the bacteria present in the digestive tract are shed [66]. As described above, factors such as vector competence, blood feeding, and infection with different microorganisms cause changes in the redox state of arthropod metabolism and, consequently, constant changes in the mosquito microbiota [25,26]. In addition to arboviruses, these changes can favor other public health harms. Hyde et al. used 16S rRNA gene sequencing to analyze larvae and adults of Aedes aegypti reared in colonies, finding the presence of antibiotic-resistant bacterial OTUs, mainly beta-lactamases [67]. The fact that mosquitoes are laboratory cultured, not captured from the wild, and yet are colonized by resistant bacteria raises the question of what role *Aedes* spp. may play in increasing bacterial resistance.

Bacteria can also be used as biomarkers of infection. The presence of the genus *Serratia* spp. in *A. aegypti* is more common in DENV and CHIKV endemic regions [48]. Upon infection, there is an interaction between the bacterial P40 polypeptide and the viral prohibitin protein and the degradation of mucins attached to the intestinal membrane, reducing its protection [25]. Thus, *Serratia* spp. is thought to facilitate arbovirus infection [68,69].

### 4.2. Virus

Recent advances in metagenomic next-generation sequencing (mNGS) have also led to the discovery of a variety of RNA viruses associated with hematophagous insects [70]. The *Aedes* spp. virome is known to be formed by arboviruses and, in greater proportion, by insect-specific viruses (ISVs) [25]. ISVs are a non-taxonomic designation for viruses that appear to be restricted to insects due to their inability to replicate in vertebrate hosts. To date, ISVs have been identified as members of the families *Flaviviridae*, *Togaviridae*, *Peribunyaviridae*, *Phenuiviridae*, *Rhabdoviridae*, *Mesoniviridae*, *Tymoviridae*, *Birnaviridae*, *Nodaviridae*, *Reoviridae*, *Parvoviridae*, *Iridoviridae*, *Permutotetraviridae*, *Iflaviridae*, *Orthomyxoviridae*, and *Totiviridae* [71,72,73]. Because of their natural association with arthropods, these viruses can be considered exclusive members of viral communities of insects (virome) and are not considered pathogens [25]. In fact, studies have shown that ISVs of adult *A. aegypti* and *A. albopictus* may play a critical role in regulating viral invasion by generating resistance to infection against other arboviruses [74].

ISVS coexist with mosquitoes for long periods of time and are passed to offspring through vertical transmission. Recent research shows similar ancestral genetic traits between ISVs and arboviruses, as well as the presence of viral RNA in the transcriptome of *Aedes* spp. mosquitoes, which may mean that the mosquito plays a role in the mutation and evolution of ISVs to arboviruses [25]. Metagenomic studies show a high prevalence of the Phasi Charoen-like virus (PCLV) in *A. aegypti* from various parts of the globe, which may indicate the coevolution of the virus with the African ancestor of *Aedes* spp. [47,75]. Further investigations are needed to clarify if and how ISVs evolved from infecting only arthropods to vertebrates.

### 4.3. Fungi

Few fungal species have been identified so far as components of the microbiome of *Aedes* spp. [48]. A study developed by Zouache et al. characterized the fungal microbiota of *A. aegypti* larvae taken from their natural habitat. The main species found belonged to the order *Pleosporales*, *Trichosphaeriales*, *Eurotiale* and *Capnodiales* (phylum *Ascomycota*); and *Tremellales*, *Polyporales* and *Wallemiales* (phylum *Basidiomycota*) [50]. Another study by Tawidian et al. analyzed fungal communities associated with the larvae and developmental site of *A. albopictus* mosquitoes using high-throughput sequencing of the internal transcribed spacer 2 meta barcode markers. The identified species belonged to the phyla *Ascomycota* (the filamentous *Aspergillus* spp., *Beauveria* spp., *Cladosporium* spp., *Penicillium* spp. and *Trichoderma* spp., and the yeast *Candida* spp.) and yeasts of the phylum *Basidiomycota* (*Cryptococcus* spp., *Rhodosporidium* spp., *Rhodotorula* spp., and *Trichosporon* spp.), similar by 44% at the genus level to other previous studies [76]. This similarity at the phylum level between the studies may support the idea of a “core microbiome”.

The presence of eukaryotes can affect the organism in different ways. The species *P. citrinum* can inhibit the development of *A. aegypti* eggs through the production of mycotoxins [50]. *Talaromyces* spp., on the other hand, is able to suppress trypsin enzyme expression in the insect midgut, facilitating arbovirus infection [25]. Fungi can also limit bacterial growth through the production of antibiotic substances, directly influencing the microbiota [50]. However, few studies have focused on the role of fungi as parasites or components of the *Aedes* spp. microbiome, making their impact unclear.

## 5. The Role of the Microbiome on Vector Competence

The effects of the microbiome on vector competence are complex and not fully understood. To date, three main hypotheses of microbiota-driven mechanisms for modulating *Aedes* spp. vector competence are highlighted: (1) innate immune system basal activation [45,46]; (2) direct antiviral activity [77]; and (3) resource competition [78]. Figure 2 represents each of these mechanisms separately. Thus, we hereby discuss the main findings regarding these possibilities.

### 5.1. Innate Immune System Activation

Xi et al. performed an interesting comparison between midgut samples of conventionally bred mosquitoes and aseptic mosquitoes in which the microbiome was depleted. These authors found that aseptic *A. aegypti* demonstrated lower levels of several Toll pathway-regulated AMPs (defensin, cecropin, attacin, and gambacin) and had higher midgut DENV2 titers. These findings support the hypothesis that the mosquito microbiota stimulates a certain basal level of immune gene expression through the activation of the Toll pathway, which, in turn, mediates the antiviral activity [38]. Another study has accordingly suggested that *Wolbachia* induces reactive oxygen species (ROS)-dependent activation of the Toll pathway in *A. aegypti*, which also leads to the production of AMPs that inhibit DENV-2 [79]. This reinforces the idea that *Wolbachia* is capable of inhibiting arbovirus infection of *Aedes* spp. It, therefore, highlights the potential of using the microbiome as a vector control strategy, which can fundamentally change the course of the fight against vector-borne diseases [79,80]. Finally, a study conducted by Ramirez et al. showed the reintroduction of bacterial isolates via a sugar meal into the midgut of *A. aegypti* mosquitoes resulted in a significant reduction in DENV infection in the case of one bacterial isolate, *Proteus* spp. Prsp_P. These authors emphasize the hypothesis that these bacteria may either indirectly exert an anti-arbovirus effect by boosting basal immunity [45]. Likewise, Terradas et al. showed that while *Wolbachia* infection induces genes in the Toll, JAK/STAT, and RNAi pathways, only reduced expression of RNAi leads to a rebound of dengue virus loads in *A. aegypti-Wolbachia*-infected cells [81]. However, the magnitude of the effect explained less than 10% of the total DENV load. In this sense, the authors found that these bacteria, by inducing microRNA (miRNA) production, suppress the expression of essential genes during viral genome methylation [81]. Their important analysis sheds light on the mosquito as a holobiont unit, in which the mosquito, its midgut microflora, and DENV are involved in complex reciprocal tripartite interactions [45,81].

### 5.2. Direct Antiviral Activity

Immune modulation is not the only mechanism through which *Wolbachia* influences the vector competence of *Aedes* spp. A study by Rainey et al. might suggest that *Wolbachia*-mediated pathogen blocking occurs early in infection before the inducible immune responses’ onset. It is currently accepted that these bacteria may block viral genome replication early in infection without a transcriptional response by endosymbiont or host small RNA pathways [82]. Concurrently, other reports suggested that *Chromobacterium* Csp_P reduces DENV infection in vector mosquitoes and has entomopathogenic and in vitro anti-pathogenic activities, mainly through Csp_P—produced stable bioactive factors with transmission-blocking and therapeutic potential. These properties highlighted its potential for the development of arbovirus control strategies [83]. Lastly, Joyce et al. also demonstrated that isolated from the *A. albopictus* midgut—namely, *Pseudomonas rhodesiae*, *Enterobacter ludwigii*, and *Vagococcus salmoninarium*—have been shown to directly inhibit the La Crosse virus independently of the mosquito immune system [84]. All these findings support the possibility that the microbiota exerts an important direct antiviral effect, mainly through the secretion of antiviral components.

### 5.3. Resource Competition

Lu et al. recently performed a study to understand why *Wolbachia* induces DENV resistance in transinfected *A. aegypti* mosquitoes, but not in *A. albopictus*. Apparently, *Wolbachia* density in the midgut, fat body, and salivary gland of *A. albopictus* is 80-, 18-, and 24-fold less than that of *A. aegypti*, respectively [78]. These results suggest that these bacteria induce density-dependent inhibition of DENV in mosquito cells, which could correlate to *Wolbachia*-arbovirus competition for limited resources. For example, intracellular cholesterol and amino acids are required by both viruses and *Wolbachia*. Thereby, the competition between them could lead to metabolite depletion, cellular stress, and subsequent inhibition of arbovirus replication—as observed in *Wolbachia*-DENV and *Wolbachia*-CHIKV co-infections [85,86,87,88].

## 6. Conclusions

Understanding the mechanisms underlying vector competence is an important area for the development of effective strategies to control the outbreak of arboviral epidemic diseases. In this study, we argue for the central role of the microbiome in modulating the intrinsic ability of a mosquito population to become infected with a particular pathogen and transmit it to susceptible hosts. Although powerful sequencing and metagenomic analyses have advanced, our understanding of the composition and functionality of the microbiota of *Aedes* spp. and our knowledge of its direct impact on vector competence remains limited. Three main hypotheses for these microbiota-driven mechanisms are currently highlighted: (1) innate immune system basal activation; (2) direct antiviral activity; and (3) resource competition. However, to date, no study has been able to identify the diverse microbiota profiles in *Aedes* spp. populations and correlate them with vector competence. The individual mechanisms of the microbiome components for basal activation of the mosquito immune system or direct antiviral activity have also not been fully elucidated. These limitations preclude the establishment of definitive conclusions. On the other hand, innovative high-throughput sequencing methods, such as RNA sequencing (RNA-seq), are now widely used in the transcriptomics field for applications like gene expression profiling. Alongside the advancing of our knowledge in microbiomics, the potential discovery of vector-related hub genes and transcripts could provide us with new insights into the tripartite interaction between *Aedes* spp. midgut microbiota, innate immune signaling pathways, and arboviruses. Future studies should focus on answering these questions.

## Figures and Tables

**Figure 1 viruses-15-00779-f001:**
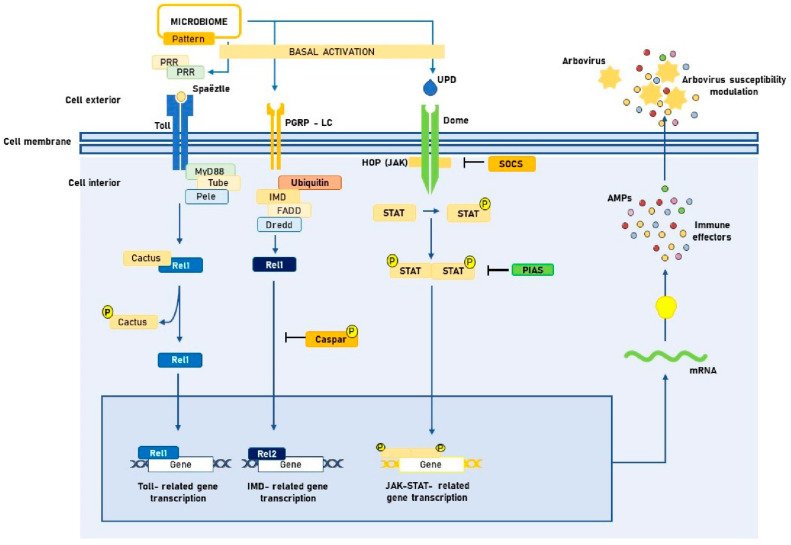
Tripartite interaction between *Aedes* spp. midgut microbiota, innate immune signaling pathways and arboviruses. PRR: Pattern recognition receptors. PAMP: Pathogen-associated molecular patterns. JAK: Janus kinase. STAT: Signal transducers and activators of transcription. UPD: Unpaired peptide ligand. PGRP-LC: Peptidoglycan recognition proteins. AMP: Antimicrobial peptide. FADD: Fas-associated death domain. SOCS: Suppressors of cytokine signaling. PIAS: Protein inhibitor of activated STAT. IMD: Immune deficiency. Created with Microsoft PowerPoint, version 16.16.24.

**Figure 2 viruses-15-00779-f002:**
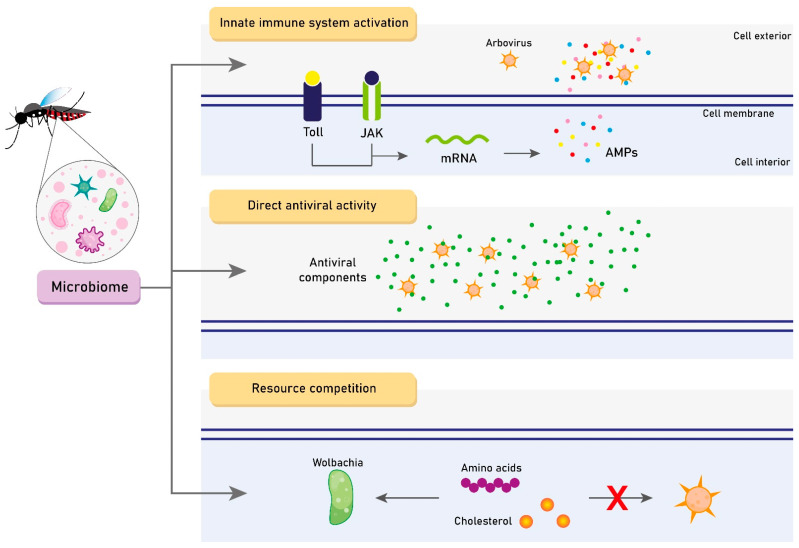
The three main hypotheses of microbiome-driven mechanisms for modulating *Aedes* spp. vector competence. JAK: Janus kinase. AMP: Antimicrobial peptide. Created with Adobe Illustrator.

**Table 1 viruses-15-00779-t001:** Findings of the studies on the microbiome composition of *Aedes* spp.

Study	Aim	Methods	Summary of Findings
do Nascimento et al. (2022) [55]	Describe the culture-dependent native microbiota associated with the female *A. aegypti* (strain PP-Campos).	Culture-dependent 16S rRNA gene sequencing	This study reveals eleven isolates from the native bacterial community of *A. aegypti* (strain PP-Campos)—*Acinetobacter* spp., *Aeromonas* spp., *Cedecea* spp., *Cellulosimicrobium* spp., *Elizabethkingia* spp., *Enterobacter* spp., *Lysinibacillus* spp., *Pantoea* spp., *Pseudomonas* spp., *Serratia* spp., and *Staphylococcus* spp. The authors also suggested that the co-infection ZIKV/*Lysinibacillus* negatively affects vector competence.
Zouache et al. (2022) [50]	Characterize the fungal microbiota of *A. aegypti* larvae taken from their natural habitat.	High-throughput sequencing	The authors identified that the entire fungal population belonged to phyla *Ascomycota* and *Basidiomycota*.
Lin et al. (2021) [56]	Compare the microbial composition of the midgut and entire bodies of *A. aegypti* and *A. albopictus* reared under the same laboratory conditions.	16S rRNA gene sequencing	*Aedes aegypti* and *A. albopictus* reared in the same laboratory harbor a similar gut bacterial microbiome but different entire body microbiota. Therefore, the gut microbiota of adult mosquitoes appears to be environmentally determined independently of host genotype, but the entire body microbiota is more genetically determined.
Díaz et al. (2021) [53]	Analyze the reproductive tract of *A. aegypti* and *A. albopictus* females.	16S rRNA gene sequencing	The authors found that blood feeding and mating cause changes in the relative populations of the microbiome components in order to favor the metabolic pathways associated with egg laying.
Hegde et al. (2019) [57]	Expand the understanding of the forces that shape the bacterial microbiome of mosquitoes (*A. aegypti*, *A. albopictus*, and *Cx. quinquefasciatus*).	16S rRNA gene sequencing	The authors identified specific bacteria that varied in abundance between mosquito species. Their analysis also demonstrated that microbial interactions affect microbiome composition and abundance of specific bacterial taxa.
Wang et al. (2018) [52]	Determine bacterial microbiota’s diversity and stability in larval habitats of the Asian tiger mosquito, *A. albopictus*, and the impact of microbiota on larval mosquito development.	16S rRNA gene sequencing	This study demonstrated that the microbiota of *A. albopictus* is strongly determined by its larval stage habitat and physiological status (e.g., developmental stage). Among all the shared bacterial genera in *A. albopictus* larvae and adults, the authors identified *Wolbachia* spp. as the dominant genus, especially in adults.
Villegas et al. (2018) [58]	Explore the potential effect that ZIKV exerts on the dynamic bacterial community harbored by the main mosquito vector *A. aegypti*.	16S rRNA gene sequencing	The authors showed that bacterial symbionts act as biomarkers of the insect’s physiological states and how they respond as a community when ZIKV invades *A. aegypti*. A core microbiota and exclusive bacterial taxa were identified, with 40 Gram-negative and 9 Gram-positive families. Two f-OTUs appeared as potential biomarkers of ZIKV infection: *Rhodobacteraceae* and *Desulfuromonadaceae*.
Rosso et al. (2018) [59]	Compare the microbiota of *A. albopictus* collected in Italy with those reported in populations from France and Vietnam.	16S rRNA gene sequencing	In this study, *A. albopictus* collected in Italy had a lower richness and a different composition of microbiota compared to samples collected in France and Vietnam. It also showed a core microbiota formed mainly of the genus *Pseudomonas* spp. Moreover, the authors found that *A. albopictus* had a 2.5% prevalence of *Wolbachia* spp. and 0.07% of *Asaia* spp.
Mancini et al. (2018) [60]	Characterize bacterial microbiota associated with the gut, salivary glands, and reproductive organs of the main representative mosquito genera in terms of geographical diffusion and public health interest.	16S rRNA gene sequencing	This study identified a shared core microbiota between different mosquito species, although interesting inter- and intra-species differences were detected. Additionally, their results showed deep divergences between genera, underlining microbiota specificity and adaptation to their host.
Dickson et al. (2017) [51]	Identify if habitat-related differences in bacterial communities in larval development sites could mediate environmental variation in vector-borne pathogen transmission.	16S rRNA gene sequencing	The authors demonstrated that exposure to different bacteria during larval development could influence the variation in adult traits in the holometabolous insect *A. aegypti*. They also showed that experimental exposure to different natural bacterial isolates at the larval stage could influence the potential transmission of medically relevant human pathogens.
Audsley et al. (2018) [61]	Assess whether there are *Wolbachia*-microbiome interactions that may affect *Wolbachia*-mediated pathogen blocking.	16S rRNA gene sequencing	This study showed that stable infection with *Wolbachia* strain wMel produces few effects on the microbiome of laboratory-reared *A. aegypti*. Although antibiotic treatment induced a measurable alteration in the microbiome composition, it did not affect DENV blocking by *w*Mel. Thus, the authors conclude that *Wolbachia*-mediated DENV blocking does not appear to rely on specific microbiome composition.
Audsley et al. (2017) [62]	Assess the impact of *Wolbachia* spp. infection of *A. aegypti* on the microbiome of wild mosquito populations (adults and larvae) collected from release sites in Cairns, Australia.	16S rRNA gene sequencing	This study showed that *Wolbachia* spp. does not substantially alter the diversity of the microbiota in mosquitoes and has the largest effects on the relative abundance of taxa that comprise a small proportion of the adult *A. aegypti* microbiome.
David et al. (2016) [28]	Profile the midgut microbial diversity throughout the *A. aegypti* lifespan and address possible determinants of microbiota structure in adult females.	Culture-dependent and -independent 16S rRNA gene sequencing	This study suggested that *A. aegypti* harbors lifelong stable core microbiota. This core was formed mainly of bacteria belonging to the genera *Pseudomonas* spp., *Acinetobacter* spp., *Aeromonas* spp., and *Stenotrophomonas* spp. and to the families *Oxalobacteraceae*, *Enterobacteriaceae*, and *Comamonadaceae*. Both dietary regime and age were found to be associated with the abundance of some bacterial groups in the *A. aegypti* microbiota.
Minard et al. (2014) [63]	Survey the bacterial diversity associated with *A. albopictus* mosquitoes by pyrosequencing 16S rDNA genes.	16S rRNA gene sequencing	The authors highlight that all mosquitoes collected from different sites have a bacterial microbiota dominated by a single taxon, *Wolbachia pipientis*, which accounted for about 99% of all 92,615 sequences obtained. Thirty-one bacterial taxa other than *Wolbachia* were identified at the genus level using different method variations. The diversity index values probably underestimated the true diversity due to the high abundance of *Wolbachia* sequences.
Coon et al. (2014) [64]	Characterize the bacterial communities of three mosquito species reared under identical conditions.	16S rRNA gene sequencing	This study demonstrated that each mosquito species contains a simple bacterial community, with *An. gambiae* and *A. aegypti* being more similar to one another than to *Ge. Atropalpus*.

## Data Availability

No new data were created or analyzed in this study. Data sharing is not applicable to this article.

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
