# Peer review of "Role of the Microbiome in Aedes spp. Vector Competence: What Do We Know?"

_viruses, 2023, doi:10.3390/v15030779_

Round 1

Reviewer 1 Report

The review “Microbiome as an influence on arbovirus vector competence: a review” discusses and summarizes the mosquito microbiome and its relation to the vector competence of mosquitos to transmit arboviruses. It also gives a good review of interactions between the mosquito’s immune pathways, microbiome and the environment. It is a relevant subject and this review contributes by summarizing the knowledge that could help the development of vector control strategies. I recommend some minor alterations to make the text clearer.

Minor

 “However, epidemics and endemics in tropical countries represent a public health hazard responsible for deaths and sequelae.” -  sentence not clear

“However, it has complications such as the demyelinating polyneuropathy Guillain-Barré Syndrome and teratogenicity” – add the name of the teratogenic syndrome “Congenital Zika Syndrome”

“Chikungunya is an acute infection whose main symptoms are high fever and severe polyarthralgia. It has a mortality rate of around 0.1%, and the elderly population is the most affected.” – One of the main public health concerns in CHIKV is chronic polyarthralgia. That should be mentioned.

“Currently, there are licensed vaccines only for dengue, yellow fever, and Japanese encephalitis virus. A CHIKV vaccine is in phase 3 and has shown good results, with a 98.5% seroconversion rate [5].” – a wrong reference and missing some reference for the statement. All references must be reviewed.

The mosquito immune modulation is well described and its relation to virus infection. What is not clear is the microbiome species found in the mosquitos that could be related to the virus infection/transmission. Is there any study that shows a group of bacteria associated with this?  That should be added in topic 2.1 making clearer the species that compromise the vector competence of mosquitos to transmit arboviruses.

Author Response

Please allow me to express my sincere gratitude for your valuable and pertinent suggestions.

In order to make our manuscript clearer and more objective, we followed all the suggestions of the reviews and focus only on Aedes spp., which caused significant changes in the structure and content of the paragraphs. First, we adapt the manuscript to an Opinion and change the title to “Role of the microbiome in Aedes spp. vector competence: what do we know?”. We asked a colleague whose native language is English has reviewed our manuscript for clarity. Unfortunately, we could not provide a language certificate by professional English language editing companies because of the scarce financial resources for research in Brazil nowadays. 

Please see the attachment with the point-by-point response. 

Reviewer 2 Report

Comments were added to the revised manuscript attachment.

This manuscript needs to be revised for proper use of English. I would recommend that it gets revised by a third party (expert in field and English speaker) for proper use of the English language. There are multiple issues with sentence structure and clarity of information. Also, the article lacks focus and structure on the topic. For instance, it provides descriptions of the bacterial flora from only few studies referencing only Aedes agypti and Ae. albopictus but no information on studies looking at viruses or protists, which are also known to be part of the mosquito microbiome (mosquito Flaviviruses?). Maybe the title should only refer to only those two mosquito species?

The manuscript's structure and focus should be strongly reconsidered, reviewed, and re-written, especially for depth and focus of topic. The manuscript does not offer a significant contribution to the field as it is because is not thorough and organized well, in addition to the language deficiencies. It does have a lot of potential but it still needs significant work.

Author Response

(The authors gave the same response as above.)

Round 2

Reviewer 2 Report

The authors did a great job in re-working the manuscript and following reviewer's advice. The quality of the manuscript improved significantly. It is well organized and uses clear English language. However, I was able to identify minor issues throughout most of the paper that are easy to correct by the authors and do not pose a major threat to acceptance for publication.

Author Response

Please allow me to express my sincere gratitude for your valuable and pertinent suggestions. We resolved all issues in the manuscript based on the peer-review report and here we make a point-by-point response to each of the issues raised in their report. In addition, as requested by the editor, we added some information that we consider relevant to the work, as well as a table and another figure that will help in the understanding of exposed points. Thank you for your time. 

______________________

  • Reviewer: Consider using infection instead of contamination, it is a more appropriate epidemiological/biological term.

Reply: Done.

  •  Reviewer: After mentioning a genus name for the first time, you can abbreviate it for the rest of the article; At the beginning of a sentence, the genus name needs to be fiully spelled out (no abbreviations at the beginning of a sentence).

Reply: Thank you. We have changed the word "Aedes" to the abbreviated form "Ae." throughout the text, except at the beginning of sentences.

  • Reviewer: does not need to be italicized.

Reply: Done.

  •  Reviewer: Spacing between the paragraph above and the one below is different.

Reply: Thank you. We proofread and re-formatted all the formatting according to the journal's standards.

  • Reviewer: What do you mean with Aedes aegypti was decentralized? (this statement is unclear)

Reply: We wanted to point out that the cited study did not show dominance of any specific bacterial genus. We have changed the sentence to "No microbe was absolutely dominant" to make it clearer.

  • Reviewer: In opposition to what?

Reply: Since we made some changes to the article, adding new information, some sentences were removed, including this one. However, our objective with this connective was to show that although some studies point to a few bacterial genera in common, another study showed that there is a diversity of mosquitoes collected from different geographical regions.

  • Reviewer: what does this side represent? (cell exterior?); what does this side represent? (cell membrane?); what does this side represent? (interior of cell?)

Reply: We added the description in the image. Thank you for your comment.

  • Reviewer: Statement needs to be explained. Do you mean that Wolbachia could be potentially used as a vector control strategy? (need a bit of follow up clarification)

Reply: Yes, we would like to mention Wolbachia as a possible strategy for arbovirosis control. To clarify, we changed the sentence to "This reinforces the idea that Wolbachia is capable of inhibiting arbovirus infection of Aedes spp. It, therefore, highlights the potential of using the microbiome as a vector control strategy, which can fundamentally change the course of the fight against vector-borne diseases.".

  • Reviewer: Need to complete the statement: do you mean "independent of the mosquito immune system?

Reply: Yes. We have added this. Thank you.
